# Automatic Grading of Individual Knee Osteoarthritis Features in Plain Radiographs Using Deep Convolutional Neural Networks

**DOI:** 10.3390/diagnostics10110932

**Published:** 2020-11-10

**Authors:** Aleksei Tiulpin, Simo Saarakkala

**Affiliations:** 1Research Unit of Medical Imaging, Physics and Technology, University of Oulu, 90220 Oulu, Finland; simo.saarakkala@oulu.fi; 2Department of Diagnostic Radiology, Oulu University Hospital, 90220 Oulu, Finland; 3Ailean Technologies Oy, 90230 Oulu, Finland

**Keywords:** multi-task learning, deep learning, transfer learning, knee osteoarthritis, OARSI grading atlas

## Abstract

Knee osteoarthritis (OA) is the most common musculoskeletal disease in the world. In primary healthcare, knee OA is diagnosed using clinical examination and radiographic assessment. Osteoarthritis Research Society International (OARSI) atlas of OA radiographic features allows performing independent assessment of knee osteophytes, joint space narrowing and other knee features. This provides a fine-grained OA severity assessment of the knee, compared to the gold standard and most commonly used Kellgren–Lawrence (KL) composite score. In this study, we developed an automatic method to predict KL and OARSI grades from knee radiographs. Our method is based on Deep Learning and leverages an ensemble of residual networks with 50 layers. We used transfer learning from ImageNet with a fine-tuning on the Osteoarthritis Initiative (OAI) dataset. An independent testing of our model was performed on the Multicenter Osteoarthritis Study (MOST) dataset. Our method yielded Cohen’s kappa coefficients of 0.82 for KL-grade and 0.79, 0.84, 0.94, 0.83, 0.84 and 0.90 for femoral osteophytes, tibial osteophytes and joint space narrowing for lateral and medial compartments, respectively. Furthermore, our method yielded area under the ROC curve of 0.98 and average precision of 0.98 for detecting the presence of radiographic OA, which is better than the current state-of-the-art.

## 1. Introduction

Osteoarthritis (OA) is the most common musculoskeletal disease leading to disability [1,2]. The etiology of OA is not currently understood, it has no cure and it eventually leads to total knee replacement [1]. Only available therapies for OA patients at the moment are behavioral interventions, e.g., weight loss, properly designed physical exercise and strengthening of joint muscles, which could lead to a temporary pain relief and decreasing OA progression rate [3].

OA is currently diagnosed using clinical examination and almost always confirmed by radiography (X-ray imaging) that is a cheap and widely used imaging modality [4]. The gold standard radiographic knee OA severity measure is the Kellgren–Lawrence (KL) grading system [5]. However, KL grade suffers from subjectivity of a practitioner and it is also a composite score not focusing separately on individual features as well as the side of OA (lateral or medial). A more recent and feature-specific approach to grade radiographic OA severity is Osteoarthritis Research Society International (OARSI) atlas [6]. Specifically, it enables grading of such features as femoral osteophytes (FO), tibial osteophytes (TO) and joint space narrowing (JSN) compartment-wise (see Figure 1). However, similar to KL score, OARSI grading suffers from subjectivity of the reader. Potentially, computer-aided methods based on Machine Learning (ML) could improve the situation by automating the OARSI grading similarly as it has been done for the KL grading [4].

Deep Learning (DL) is a state-of-the art ML approach that allows learning of features directly from the data, and it has recently revolutionized the field of medical image analysis by surpassing the conventional computer vision techniques that required manual engineering of data representation methods [7]. In the OA research field, several studies demonstrated success in the analysis of Magnetic Resonance Imaging (MRI) data [8,9], basic research [10], prediction of knee osteoarthritis progression [11] and, in particular, automation of the KL-grading of knee and hip radiographs using deep convolutional neural networks (CNN) [4,12,13,14]. However, only a few attempts have been made to assess individual knee OA features from plain radiography.

### Contributions

In this study, we present a robust DL-based multi-task framework for automatic simultaneous OARSI and KL scoring and validate it with an independent test set. The main contributions of this paper can be summarized as follows:We demonstrate a possibility to accurately predict individual knee OA features and overall knee OA severity from plain radiographs simultaneously. Our method significantly outperforms previous state-of-the-art approach [15].Compared to the previous study [15], for the first time, we utilize two independent datasets for training and testing in assessing automatic OARSI grading: OAI and MOST, respectively.We perform an extensive experimental validation of the proposed methodology using various metrics and explore the influence of network’s depth, utilization of squeeze-excitation and ResNeXt blocks [16,17] on the performance, as well as ensembling, transfer learning and joint learning of KL and OARSI grading tasks.Finally, we also release the source codes and the pre-trained models allowing full reproducibility of our results.

## 2. Materials and Methods

### 2.1. Overview

In this study, we used bilateral posterior–anterior (PA) fixed-flexion knee radiographs as our training and testing material. To pre-process the data, we performed knee joint area localization using random forest regression voting [18] and applied intensity normalization. Subsequently, utilizing a transfer learning approach [19], we initialized a convolutional part of our model from an ImageNet [20] pre-trained model and predicted the KL and OARSI grades simultaneously. The overall pipeline is graphically illustrated in Figure 2. Finally, we note that all the experiments in our study were performed in accordance with relevant guidelines and regulations.

### 2.2. Data

We utilized two publicly available knee X-ray datasets: OAI (https://nda.nih.gov/oai/) and MOST (http://most.ucsf.edu). Fixed-flexion bilateral Posterior–Anterior (PA) images acquired using a Synaflexer positioning frame with the X-ray beam angle of 10° were used in both datasets [21].

OAI is a longitudinal study of 4796 participants examined with X-ray, MRI and other means during nine follow-up examinations (0–96 months). MOST is a similar dataset to OAI, but acquired from 3026 participants who were not part of OAI. MOST included four follow-up examinations with imaging (0–84 months). The main inclusion criterion for both cohorts was the presence of OA or an increased risk of developing it. The age of the subjects was of 45–79 and 50–79 years old for OAI and MOST, respectively.

Both OAI and MOST studies were approved by the institutional review board of the University of California San Francisco and the data acquisition sites. The informed consent was obtained from all the subjects participated in the study and all the data were fully anonymized. Further details regarding the ethical approvals and methodology of data acquisition can be found by the aforementioned website links.

### 2.3. Data Pre-Processing

We performed two types of data pre-processing—on the metadata and image levels. As such, we first removed the data with the missing labels from both OAI and MOST datasets. After filtering out the missing labels (KL and OARSI scorings), we derived a training set of 19704 knees from the OAI dataset and a testing set of 11743 knees from the MOST dataset. Eventually, from the OAI dataset, we used the data from all the subjects (4796) and from MOST we excluded five subjects due to missing metadata (total number of subjects—3021). We note here that for each individual subject our dataset contained one or more knee X-rays. The full description of the data is presented in Table 1. A visual representation of distribution of OARSI grades in lateral and medial compartments is presented in Appendix A, respectively. Sample availability: MOST and OAI datasets are publicly available at http://most.ucsf.edu/ and https://nda.nih.gov/oai/, respectively. Our dataset splits, source codes and the pre-trained models are publicly available: https://github.com/MIPT-Oulu/KneeOARSIGrading.

In contrast to the previous studies [4,12,15,22], we applied a different approach to localize the region of interest (ROI). Specifically, we utilized the random forest regression voting approach implemented in a BoneFinder tool [18] to localize the knee joint landmarks. Subsequently, we cropped the ROIs of 140×140 mm from the right and the left knees and rotated each individual knee image to horizontally align the tibial plateaus. We also applied histogram clipping and global contrast normalization to each localized knee joint image as proposed in [4]. Finally, we rescaled all the images to 310×310 pixels (0.45 mm resolution) using bilinear interpolation.

### 2.4. Network Architecture

Our approach is based on ensembling of two convolutional neural networks. Each model within the ensemble consists of two parts. The first part is convolutional and was pre-trained on ImageNet [20]. The second part consists of seven independent fully-connected (FC) layers each corresponding to its own task (a KL grade and six OARSI grades). To connect these two parts, we utilized an average pooling layer after the convolutional block of our network.

For the convolutional part of the model, we evaluated various network backbones from Resnet family [23]. As such, we firstly utilized Resnet-18, Resnet-34 and Resnet-50 to assess whether the depth of the model plays any role in predicting OARSI and KL grades in a multi-task setting. Then, we tested the use of squeeze-excitation (SE) blocks by utilizing SE-resnet-50 model architecture [16]. Finally, we also used the blocks from ResNeXt model combined with SE modules as proposed in [16].

In addition to the experiments presented in Section 3, we also evaluated Global Weighted Average Pooling (GWAP) [24] instead of a simple average pooling and also GWAP with a hidden layer. Despite being attractive, GWAP and its modification did not lead to improvements on cross-validation. Therefore, we present only the results with the average pooling in the paper.

### 2.5. Training Strategy

Our experimental setup employed a five-fold subject-wise stratified cross-validation. At the model selection phase, we calculated Cohen’s kappa coefficients and also the balanced accuracy on out-of-fold sample, thereby utilizing the whole training set. Eventually, we selected two models that performed best in the majority of the tasks and used them in the final ensemble. At the test phase, we performed the inference for each of the model in the ensemble (five snapshots per model) and eventually averaged their predictions.

In all the experiments, the same training strategy was utilized per type of experiment (with and without transfer learning from ImageNet). Firstly, we performed the transfer learning experiments jointly training to predict both KL and OARSI grades to select the best network architectures. Secondly, we trained the same models from scratch using the random weight initialization. Thirdly, we also attempted to predict solely OARSI grades without joint training with KL grade prediction task while still using the ImageNet weights for model initialization.

We executed the transfer learning experiments as follows. For the first training epoch, only the FC layers were trained with the learning rate (LR) of 0.01. Subsequently, we unfroze the convolutional layers and trained the full network with the LR of 0.001. Finally, at the beginning of the third epoch, we switched to LR of 0.0001 and trained all the models for the remaining eighteen epochs. Adam optimizer was used in all the experiments [25].

The training of all the models was regularized using data augmentations from SOLT library [26]. We used random cropping of 300×300 pixels, Gaussian noise addition and random gamma correction. Besides data augmentations, we also used weight decay of 0.0001 and dropout of 0.5 (inserted before each FC layer). PyTorch v1.0 was used to train all the models [27].

The training of the models from scratch was done with exactly the same hyper-parameters as in the transfer learning experiments besides the starting LR and the LR schedule. As such, the starting LR was set to 0.0001 and it was dropped ten times after the 10th and 15th epochs.

Finally, it is worth noting that, due to data imbalance, we tested various weighted data sampling strategies (e.g., balancing the KL grade distribution as in [4]). However, they did not lead to improvement in the scores.

## 3. Results

### 3.1. Cross-Validation Results and Backbone Selection

We performed a thorough evaluation of Resnet-18, Resnet-34, Resnet-50, SE-Resnet-50 and SE-Resnet-50-32x4d (SE-Resnet-50 with ResNext blocks) using cross-validation (see Table 2 and Appendix A). Based on cross-validation, we selected two models for further investigation: SE-Resnet-50 and SE-Resnet-50-32x4d. In particular, we investigated the added value of jointly training OARSI and KL grading tasks, added value of transfer learning and, finally, model ensembling. Our experiments indicate that jointly training KL and OARSI grading tasks hurts the performance of automatic OARSI grading. Besides, we found that transfer learning helps significantly for the model convergence. Finally, ensembling the two best models allowed increasing the performance in both tasks. Further, we report the results for the ensemble of SE-Resnet-50 and SE-Resnet-50-32x4d as our final model since it yielded the best performance in terms of both Cohen’s kappa and balanced accuracy (see the latter in Appendix A).

### 3.2. Test-Set Performance

Based on the cross-validation, we selected our final ensemble model to be evaluated on the test set. Its test set performance and also the current state-of-the-art performance reported previously by Antony et al. are presented in Table 3. Our method yielded Cohen’s kappa of 0.82 (0.82–0.83) and balanced accuracy of 66.68% (0.66–0.67%) for KL grading. For OARSI grading tasks, the developed method yielded Cohen’s kappa and balanced accuracy of 0.79 (0.78–0.80) and 63.58% (62.46–64.84%), 0.84 (0.84–0.85) and 68.85% (68.03–69.61%), 0.94 (0.93–0.95) and 78.55% (76.70–80.31%), 0.84 (0.83–0.85) and 65.49% (64.49–66.47%), 0.83 (0.83–0.84) and 72.02% (70.99–0.72.96%) and 0.90 (0.89–0.90) and 80.66% (79.82–81.54%) for TO, FO and JSN in lateral and medial compartments, respectively. The 95% confidence intervals here were computed using stratified bootstrapping with 500 iterations.

Besides the metric-based evaluation, we also analyzed the confusion matrices for both OARSI and KL grades, as well as the performance of detecting OA, osteophytes presence and abnormal JSN in each knee joint compartment (Figure 3). The confusion matrices for the OARSI grades are presented in Figure 4. The confusion matrix for the KL grading is presented in Appendix A.

### 3.3. Evaluation on the First Follow-Up of MOST Dataset

To verify the impact of repeated subjects in the test set, we made an additional evaluation of the models using only the data from the first imaging follow-up from the MOST dataset. We obtained the Cohen’s kappa values of 0.83 (0.82–0.84), 0.79 (0.77–0.81), 0.84 (0.82–0.85), 0.94 (0.93–0.95), 0.86 (0.84–0.87), 0.83 (0.82–0.84) and 0.91 (0.90–0.91) for KL as well as for OARSI grades (FO, TO and JSN for lateral and medial compartments), respectively.

The balanced accuracy in KL and OARSI grading tasks were of 67.90% (66.57–69.13%), 64.72% (62.23–67.16%), 69.11% (66.99–70.99%), 80% (76.48–83.21%), 65.80% (63.68–67.74%), 72.51% (70.40–74.46%) and 83.34% (81.94–84.64%). Here, the 95% confidence intervals were computed via stratified bootstrapping with 500 iterations.

### 3.4. Evaluation of Performance with Respect to the Stage of OA

In addition to the results presented on the whole MOST dataset, we performed additional evaluations on its three strata, grouping KL0 and KL1 into a “No OA” group, KL2 in “Early OA” group and KL3 and KL4 into “Severe OA” group. These results are shown in Table 4.

In addition to the metrics shown in Table 3, we also computed the F1 score performance weighing by the support as well as the macro-average. The former allows judging the performance of model when detecting both positive and negative samples. The latter calculates the average of F1 scores computed for each of the classes.

## 4. Discussion

In this study, we developed a DL-based method to perform an automatic simultaneous OARSI and KL grading from knee radiographs using transfer learning. The developed approach employed two deep residual networks with 50 layers that incorporated SE and ResNeXt blocks [16]. Compared to the previous state-of-the-art [4,15], our model performs significantly better in simultaneous OARSI and KL grading as well as in the detection of radiographic OA presence. The agreement of the predicted OARSI grades on the test set with the test labels exceeds both previously reported human [15,28] and algorithm [15] performances (see Table 3). Moreover, this is the first study in OA when an independent test set was used for automatic OARSI grading from plain radiographs.

To the best of our knowledge, Oka et al. [29] were the first to report automatic analysis of individual knee OA features. Later, Thomson et al. (2016) [30] used a more robust setup and an advanced methodology based on the shape and texture descriptors to evaluate the presence of osteophytes and radiographic OA (KL ≥2). The authors reported the area under the receiver operating characteristic (ROC) curve for detecting osteophytes as 0.85. That study, however, had two main limitations. Firstly, the test set size was relatively small compared to the other OA studies [4,12]. Secondly, the problem of binary discrimination between osteophytes of OARSI Grades 0–1 and 2–3 may not be clinically relevant as Grade 1 already indicates the presence of an osteophyte [6].

In contrast to those studies, the above-mentioned limitations were addressed in the recent study by Antony [15] where a CNN-based approach for simultaneous analysis of KL and OARSI grades was proposed. However, the limitation of that study was a dataset that consisted of a combination of MOST (Multi-center Osteoarthritis study) and OAI (Osteoarthritis Initiative) data and, furthermore, the agreements between the method’s predictions and the test set labels were shown to be lower than inter-rater agreements between the human observers for KL and OARSI grades. Here, we tackle both of these limitations and demonstrate an excellent agreement of our method with the test set labels.

Other related works to this study are by Antony et al. [12,31] and Tiulpin et al. [4]. While the studies by Antony et al. were pioneering in the field, the study by Tiulpin et al. produced the new state-of-the-art results in KL grading—Cohen’s quadratic kappa of 0.83—as well as in radiographic OA detection—area under the ROC curve of 0.93. The balanced accuracy was 66.71%.

We conducted our experiments in multiple settings: joint training for predicting OARSI and KL grades with and without transfer learning and also prediction of solely OARSI grades without the use of transfer learning. Our results on cross-validation indicate that transfer learning is useful for automatic OARSI grading and also that joint prediction of KL and OARSI grades leads to worse performance. However, the latter is a clinically relevant setting since the KL grade allows for a composite assessment of the knee condition and it is used by practitioners world-wide, in contrast to the OARSI grades. However, OARSI grades allow for evaluation of individual knee features and can be utilized for more comprehensive quantification of OA-related changes between the follow-up examinations when monitoring the OA progression in time. Therefore, despite worse performance, joint prediction of KL and OARSI grades has additional clinical value. To overcome the limitations of learning joint KL and OARSI tasks, we performed an ensembling of two models selected using cross-validation—SE-Resnet-50 and SE-ResNext50-32x4d. Our results indicate the notable improvement on cross-validation compared to all the investigated single models (Table 2 and Appendix A).

This study, while providing the new state-of-the-art results in automatic OARSI grading and detection of radiographic OA presence, still has some limitations. Firstly, compared to the previous work [4], we did not analyze the attention maps produced by our method. Attention maps could provide further insights into the specific decisions made by the CNN [32]. However, in this study, we decided to mainly focus on a large-scale experimental evaluation of the conventional transfer learning rather than on model interpretation. Secondly, the presented ensemble approach is computationally heavy due to ensembling and, hypothetically, could affect the real-life use of the developed method unless the model is deployed on GPU. Potentially, techniques such as knowledge distillation [33] could help to decrease the computational effort needed for model execution. Thirdly, we utilized the whole knee images for training our models. Future studies should compare this approach with the Siamese model proposed by Tiulpin et al. [4]. Fourthly, we considered only the OARSI grades that had sufficient amount of training and test data. Therefore, some additional OARSI features (medial tibial attrition, medial tibial sclerosis and lateral femoral sclerosis) were not considered at all, which could be the target of future studies. Finally, our test set included the data from the same patients obtained from multiple follow-ups. However, this should not significantly affect our results and rather made them less optimistic due to the fact that MOST is a cohort of subjects at risk that have progressing osteoarthritis. Therefore, the appearance of the images changes across the follow-ups and the overall dataset still contains diverse images. To verify the significance of the this limitation, we made an additional evaluation of the models using only the data from the first imaging follow-up from MOST dataset (Section 3.4).

The final and the main limitation of this work is a possible bias in performance of the algorithm. While the method performs well on the whole dataset, it still lacks good results when we stratify the test set according to the stages of OA. Specifically, while the reported results on the whole MOST dataset are in line or better than the previous state-of-the-art [4,15], one can observe that the F1 score (both weighted and macro-averaged) improve when the algorithm is tested on population with severe OA (see Table 4). Therefore, the future studies need to put a bigger emphasis on improving the scores on cases when OA is not severe.

To conclude, this study demonstrated the first large-scale experiment for automatic KL and OARSI grading. Despite the limitations, we believe that the developed methodology has potential to become a useful tool in clinical OA trials and also could provide better quantitative information about the knees of the patients who already have OA for a clinician in a systematic manner.

## Figures and Tables

**Figure 1 diagnostics-10-00932-f001:**
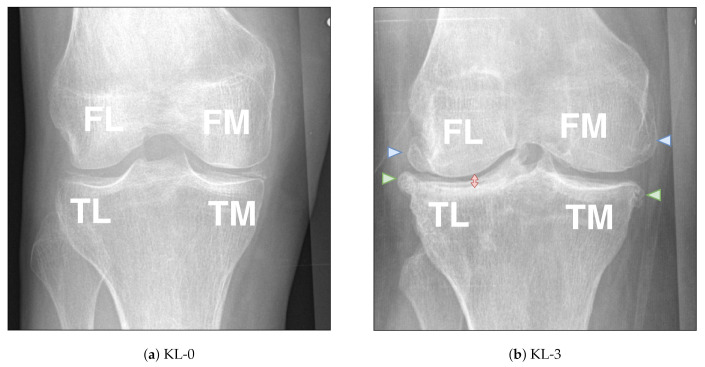
Examples of knee osteoarthritis features graded according to the Osteoarthritis Research Society (OARSI) grading atlas and Kellgren–Lawrence (KL) grading scale. FL, TL, FM and TM represent the femoral lateral, tibial lateral, femoral medial and tibial medial compartments, respectively. I (**a**) A right knee without visual OA-related changes is presented (KL 0, all OARSI grades also zero). (**b**) An image of a right knee with severe OA (KL 3) is presented. Blue triangles highlight the osteophytes in femur and the green triangles highlight the osteophytes in tibia. Red arrow highlights the joint-space narrowing (JSN). Here, the osteophytes for FL, TL, FM and TM compartments are all of Grade 3. JSN in the lateral compartment is of Grade 2 and in the medial compartment it is of Grade 0.

**Figure 2 diagnostics-10-00932-f002:**
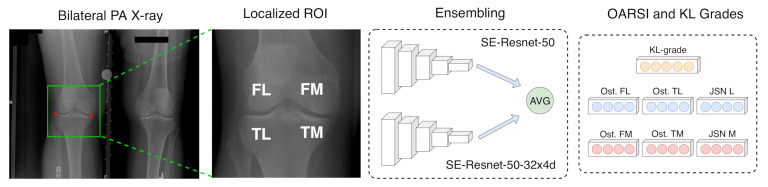
Schematic representation of the workflow of our approach. We used transfer learning from ImageNet and trained two deep neural network models, averaged their predictions and predicted totally six knee joint radiographic features according to the OARSI grading atlas as well as a the KL grade. OARSI grades for osteophytes in femoral lateral (FL), tibial-lateral (TL), femoral-medial (FM) and tibial-medial (TM) compartments as well as the joint space narrowing (JSN) grades in lateral and medial compartments were predicted.

**Figure 3 diagnostics-10-00932-f003:**
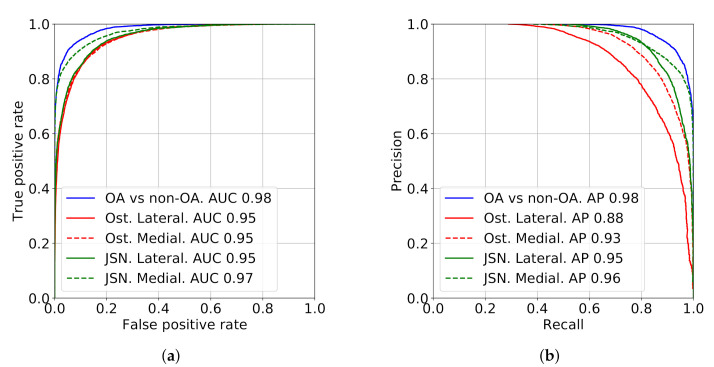
ROC and precision-recall curves demonstrating the performance of detecting the presence of radiographic OA (KL ≥2) osteophytes (grade ≥1) and joint-space narrowing (grade ≥1).

**Figure 4 diagnostics-10-00932-f004:**
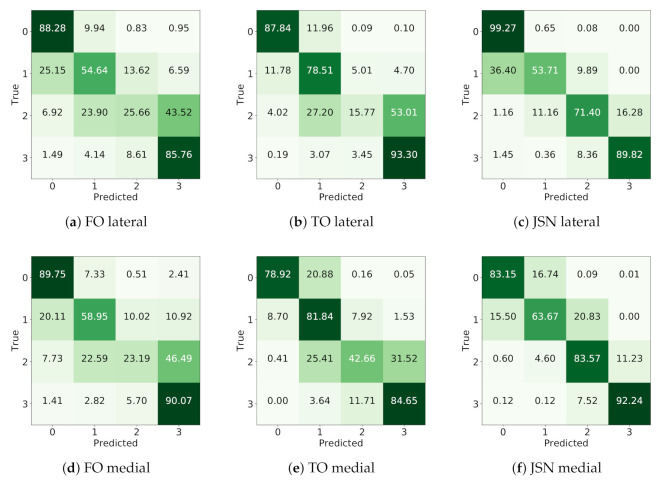
Confusion matrices for the OARSI grades prediction tasks: (**a**–**c**) the matrices for femoral osteophytes (FO), tibial osteophytes (TO) and joint space narrowing (JSN) automatic grading in lateral compartment, respectively; and (**d**–**f**) the confusion matrices in the same order, but for the lateral compartment. The numbers indicate percentages.

**Table 1 diagnostics-10-00932-t001:** Description of the datasets used in this study. We used all the follow-up examinations from Osteoarthritis Initiative (OAI) and Multi-Center Osteoarthritis Study (MOST). L and M indicate lateral and medial compartments, FO and TO indicate femoral and tibial osteophytes and JSN indicates joint space narrowing. KL indicates the Kellgren–Lawrence grade. # sign stands for the amount of data.

Dataset	# Images	Grade	# KL	# FO		# TO		# JSN	
				L	M	L	M	L	M
OAI (Train)	19704	0	2434	11,567	10,085	11,894	6960	17,044	9234
		1	2632	4698	4453	5167	9181	1160	5765
		2	8538	1748	2068	1169	2112	1061	3735
		3	4698	1691	3098	1474	1451	439	970
		4	1402	-	-	-	-	-	-
MOST (Test)	11743	0	4899	9008	7968	8596	6441	10,593	7418
		1	1922	1336	1218	1978	3458	465	1865
		2	1838	795	996	647	1212	442	1721
		3	2087	604	1561	522	632	243	739
		4	997	-	-	-	-	-	-

**Table 2 diagnostics-10-00932-t002:** Cross-validation results (out of fold): Cohen’s kappa coefficients for each of the trained tasks on out-of-fold sample (OAI dataset). The best results task-wise are highlighted in bold. We selected two best models for thorough evaluation: SE-Resnet-50 ${}^{†}$ and SE-ResNext50-32x4d ${}^{‡}$. We trained these models from scratch (*) and also with transfer learning, but without the KL-grade (**). Finally, in the last row, we show the results for the ensembling of these models. L and M indicate lateral and medial compartments, respectively. FO and TO indicate femoral and tibial osteophytes and JSN indicates joint space narrowing, respectively. KL indicates the Kellgren–Lawrence grade.

Backbone	KL	FO		TO		JSN	
		L	M	L	M	L	M
Resnet-18	0.81	0.71	0.78	0.80	0.76	0.91	0.87
Resnet-34	0.81	0.69	0.78	0.80	0.76	0.90	0.87
Resnet-50	0.81	0.70	0.78	0.81	0.78	0.91	0.87
SE-Resnet-50 ${}^{†}$	0.81	0.71	0.79	0.81	0.78	0.91	0.87
SE-ResNext50-32x4d ${}^{‡}$	0.81	0.72	0.79	0.82	0.78	0.91	0.87
SE-Resnet-50 *	0.78	0.66	0.73	0.76	0.70	0.91	0.87
SE-ResNext50-32x4d *	0.77	0.67	0.73	0.75	0.71	0.91	0.87
SE-Resnet-50 **	-	0.71	0.79	0.82	0.78	0.91	**0.88**
SE-ResNext50-32x4d **	-	**0.73**	**0.80**	**0.83**	0.78	0.91	**0.88**
Ensemble ${}^{†,‡}$	**0.82**	**0.73**	**0.80**	**0.83**	**0.79**	**0.92**	**0.88**

**Table 3 diagnostics-10-00932-t003:** Test set performance of our ensemble method with SE-Resnet50 and SE-ResNext50-32x4d backbones. MSE, A and K indicate the mean squared error, balanced accuracy and Cohen’s kappa, respectively. As a comparison, the three rightmost columns show the state-of-the-art (SOTA) performance reported by Antony et al. in a similar work. L and M indicate lateral and medial compartments, respectively. FO and TO indicate femoral and tibial osteophytes and JSN indicates joint space narrowing, respectively. KL indicates the Kellgren–Lawrence grade.

Side	Grade	A	K	MSE	$A_{\mathrm{SOTA}}$	$K_{\mathrm{SOTA}}$
L	FO	0.69 (0.68–0.7)	0.84 (0.84–0.85)	0.22 (0.21–0.23)	44.3	0.47
	TO	0.64 (0.62–0.65)	0.79 (0.78–0.8)	0.33 (0.31–0.34)	47.6	0.52
	JSN	0.79 (0.77–0.8)	0.94 (0.93–0.95)	0.04 (0.04–0.05)	69.1	0.80
M	FO	0.72 (0.71–0.73)	0.83 (0.83–0.84)	0.26 (0.25–0.27)	45.8	0.48
	TO	0.65 (0.64–0.67)	0.84 (0.83–0.85)	0.41 (0.38–0.44)	47.9	0.61
	JSN	0.81 (0.8–0.82)	0.9 (0.89–0.9)	0.20 (0.19–0.20)	73.4	0.75
Both	KL	0.67 (0.66–0.67)	0.82 (0.82–0.83)	0.68 (0.65–0.70)	63.6	0.69

**Table 4 diagnostics-10-00932-t004:** Test set performance of our ensemble method with SE-Resnet50 and SE-ResNext50-32x4d backbones with respect to the stage of osteoarthritis. F1, MSE, A and K indicate F1-score (geometric average of precision and recall) either weighted by the support or by averaging F1 scores across the classes (macro averaging), mean squared error, balanced accuracy and Cohen’s kappa, respectively. L and M indicate lateral and medial compartments, respectively. FO and TO indicate femoral and tibial osteophytes and JSN indicates joint space narrowing, respectively. KL indicates the Kellgren–Lawrence grade. Here, “No OA” indicates knees with KL0 and KL1, “Early OA” indicates knees with KL2 and “End stage” indicates knees with KL3 and KL4.

Stage	Side	Grade	F1 (weighted)	F1 (macro)	MSE	A	K
No	L	FO	0.94 (0.93–0.94)	0.36 (0.35–0.74)	0.08 (0.07–0.09)	0.85 (0.8–0.89)	0.47 (0.42–0.53)
		TO	0.95 (0.94–0.95)	0.31 (0.29–0.42)	0.08 (0.07–0.1)	0.74 (0.66–0.8)	0.26 (0.19–0.32)
		JSN	0.99 (0.98–0.99)	0.71 (0.62–0.8)	0.01 (0.01–0.02)	0.72 (0.61–0.84)	0.42 (0.23–0.59)
	M	FO	0.85 (0.84–0.86)	0.49 (0.48–0.5)	0.17 (0.15–0.19)	0.81 (0.78–0.83)	0.49 (0.45–0.52)
		TO	0.95 (0.95–0.96)	0.34 (0.32–0.47)	0.07 (0.06–0.09)	0.79 (0.73–0.85)	0.34 (0.26–0.41)
		JSN	0.86 (0.85–0.88)	0.46 (0.45–0.48)	0.16 (0.15–0.18)	0.8 (0.76–0.83)	0.45 (0.4–0.49)
Early	L	FO	0.94 (0.93–0.94)	0.36 (0.35–0.74)	0.08 (0.07–0.09)	0.85 (0.8–0.89)	0.47 (0.42–0.53)
		TO	0.95 (0.94–0.95)	0.31 (0.29–0.42)	0.08 (0.07–0.1)	0.74 (0.66–0.8)	0.26 (0.19–0.32)
		JSN	0.99 (0.98–0.99)	0.71 (0.62–0.8)	0.01 (0.01–0.02)	0.72 (0.61–0.84)	0.42 (0.23–0.59)
	M	FO	0.85 (0.84–0.86)	0.49 (0.48–0.5)	0.17 (0.15–0.19)	0.81 (0.78–0.83)	0.49 (0.45–0.52)
		TO	0.95 (0.95–0.96)	0.34 (0.32–0.47)	0.07 (0.06–0.09)	0.79 (0.73–0.85)	0.34 (0.26–0.41)
		JSN	0.86 (0.85–0.88)	0.46 (0.45–0.48)	0.16 (0.15–0.18)	0.8 (0.76–0.83)	0.45 (0.4–0.49)
Severe	L	FO	0.66 (0.63–0.69)	0.60 (0.57–0.63)	0.48 (0.41–0.56)	0.64 (0.61–0.67)	0.81 (0.78–0.83)
		TO	0.64 (0.6–0.66)	0.57 (0.54–0.6)	0.77 (0.65–0.89)	0.61 (0.57–0.64)	0.74 (0.7–0.77)
		JSN	0.94 (0.93–0.95)	0.66 (0.61–0.72)	0.07 (0.05–0.08)	0.68 (0.64–0.74)	0.96 (0.95–0.97)
	M	FO	0.60 (0.57–0.63)	0.6 (0.56–0.64)	0.47 (0.42–0.52)	0.62 (0.58–0.65)	0.72 (0.69–0.75)
		TO	0.64 (0.61–0.67)	0.56 (0.52–0.59)	0.85 (0.72–0.97)	0.57 (0.53–0.61)	0.66 (0.61–0.71)
		JSN	0.88 (0.86–0.9)	0.70 (0.67–0.75)	0.13 (0.11–0.16)	0.73 (0.69–0.8)	0.93 (0.92–0.94)

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
