# Peer review of "Automatic Grading of Individual Knee Osteoarthritis Features in Plain Radiographs Using Deep Convolutional Neural Networks"

_diagnostics, 2020, doi:10.3390/diagnostics10110932_

Round 1

Reviewer 1 Report

No substantial remarks on this nice work.

I would just suggest adding some briefs comments about the potential applicability of the study findings to radiological and clinical real practice.

Reviewer 2 Report

In this study, authors developed a deep learning based model for automatic grading of KL and OARSI with the AUC of 0.98 for detecting the presence of radiographic OA (KL >=2). The manuscript is well written and methodology is robust. However, there are some concerns regarding the utilized datasets and study population which should be clarified to minimize the possibility of selection bias.

-Based on the provided numbers, authors have included all available knee radiographs of all OAI subjects including all annual follow-up imaging studies. If it is true, it should be clearly mentioned in the manuscript that X numbers of XRs from Y number of subjects were included, some of the included subjects have multiple imaging in the dataset. The same for MOST.

-Please clarify the main inclusion criteria of each cohort as well.

-Regarding the OAI dataset, this cohort has 3 main parts including OA incidence, OA progression, and normal subjects. Is it the same for the MOST cohort, in terms of inclusion criteria? Study population should be clearly described in the manuscript.

-Both these cohorts mainly included subjects with or at-risk of knee OA which could drastically limit the utility of the developed algorithm in normal subjects (i.e., not end-stage knee OA). I would recommend rerunning the test analysis in different subsets of patients. Does the model have the same performance in incidence cohort vs progression cohort? The possibility of selection bias could highly impact the utility of developed algorithm.

-The manuscript is well written, but it should be rephrased in some sections.  In the last paragraph of introduction, authors described their findings and compared it with the previously published literature – transfer these parts to the discussion.

-Add a paragraph at the end of the discussion and point out the main limitations of study.
